# Propranolol Promotes Monocyte-to-Macrophage Differentiation and Enhances Macrophage Anti-Inflammatory and Antioxidant Activities by NRF2 Activation

**DOI:** 10.3390/ijms25073683

**Published:** 2024-03-26

**Authors:** Sonia Maccari, Elisabetta Profumo, Luciano Saso, Giuseppe Marano, Brigitta Buttari

**Affiliations:** 1Center for Gender Medicine, Italian National Institute of Health, 00161 Rome, Italy; sonia.maccari@iss.it (S.M.); giuseppe.marano@iss.it (G.M.); 2Department of Cardiovascular and Endocrine-Metabolic Diseases, and Aging, Italian National Institute of Health, 00161 Rome, Italy; elisabetta.profumo@iss.it; 3Department of Physiology and Pharmacology “Vittorio Erspamer”, Sapienza University, 00185 Rome, Italy; luciano.saso@uniroma1.it

**Keywords:** propranolol, β-adrenergic receptor, monocytes, macrophages, cytokines

## Abstract

Adrenergic pathways represent the main channel of communication between the nervous system and the immune system. During inflammation, blood monocytes migrate within tissue and differentiate into macrophages, which polarize to M1 or M2 macrophages with tissue-damaging or -reparative properties, respectively. This study investigates whether the β-adrenergic receptor (β-AR)-blocking drug propranolol modulates the monocyte-to-macrophage differentiation process and further influences macrophages in their polarization toward M1- and M2-like phenotypes. Six-day-human monocytes were cultured with M-CSF in the presence or absence of propranolol and then activated toward an M1 pro-inflammatory state or an M2 anti-inflammatory state. The chronic exposure of monocytes to propranolol during their differentiation into macrophages promoted the increase in the M1 marker CD16 and in the M2 markers CD206 and CD163 and peroxisome proliferator-activated receptor ɣ expression. It also increased endocytosis and the release of IL-10, whereas it reduced physiological reactive oxygen species. Exposure to the pro-inflammatory conditions of propranolol-differentiated macrophages resulted in an anti-inflammatory promoting effect. At the molecular level, propranolol upregulated the expression of the oxidative stress regulators NRF2, heme oxygenase-1 and NQO1. By contributing to regulating macrophage activities, propranolol may represent a novel anti-inflammatory and immunomodulating compound with relevant therapeutic potential in several inflammatory diseases.

## 1. Introduction

Macrophages are phagocytic innate immune cells that seed all tissues within the body that contribute to maintain homeostasis through the clearance of apoptotic cells and the production of growth factors [1]. These cells are able to remove invading pathogens through their phagocytic and antigen-presenting activities and are also the main cell type involved in the regulation of tissue remodeling/healing, cell proliferation and angiogenesis under certain microenvironmental conditions [2,3]. Plasticity is a hallmark of cells of the monocyte/macrophage lineage [4]. In general, macrophages can be activated by local cytokines and mediators to become M1 or M2 macrophages. M1 activity inhibits cell proliferation and causes tissue damage, while M2 activity promotes cell proliferation and tissue repair. However, M1 and M2 activation phenotypes represent two ends of a functional spectrum of macrophage polarization states [5]. In general, M1 macrophages are considered pro-inflammatory cells, whereas M2 macrophages are anti-inflammatory [6]. M1 macrophages secrete pro-inflammatory cytokines and are involved in the maintenance of homeostasis against infections and cancer development; M2 macrophages secrete anti-inflammatory cytokines, show strong phagocytic activity and are involved in tissue repair and immune tolerance. Inflammation is a biological defensive response put in place to protect the organism against infections and trauma. However, once activated, the inflammatory response must be controlled to prevent it from becoming chronic and causing tissue damage. In this regard, macrophages play a fundamental role. In several pathologies, M1/M2 macrophages are critical players in the disease process. Evidence exists demonstrating that macrophage polarization influences the outcome of infections, chronic inflammatory diseases and tumors, and the development of strategies to modulate this process is of great interest in the prevention and treatment of different pathologies. For example, an overbalance towards M1 macrophages can result in obesity and cardiovascular diseases [7,8]. Obesity is associated with the presence of M1 macrophages, whereas people who are thin are characterized by a prevalence of M2 macrophages that are involved in adipose tissue homeostasis maintenance, inflammation prevention and the promotion of insulin sensitivity [9]. On the other hand, an overbalance towards M2 macrophages can result in tumor growth [10]. Different M2-like macrophage subsets have been described, all of which are able to produce high levels of the anti-inflammatory cytokine IL-10 [11]. In particular, a novel M2-like subset consists of tumor-associated macrophages (TAMs) that inhibit pro-inflammatory M1 macrophages and represents the major inflammatory component of the tumoral tissue, contributing to angiogenesis and tumor metastasis [12]. Promoting the switch of TAMs from the pro-tumor M2 phenotype to an anti-tumor M1 phenotype represents a pivotal target for the immunotherapy of cancer. In general, the determination of the mechanisms underlying the multiple roles played by M1/M2 macrophages in different disease conditions represents a basis for macrophage-centered therapeutic strategies. Despite tremendous efforts, the molecular regulation of monocyte/macrophage function remains incompletely understood.

A way to regulate monocyte/macrophage functions can occur through stimulation of adrenergic receptors (ARs). Adrenergic pathways represent the main channel of communication between the nervous system and the immune system [13]. This cross-talk is required to maintain and restore homeostasis [14]. The main ARs expressed in innate immune cells are of the β1 and β2 subtypes, which are known to influence cell inflammatory response. The functional consequences of β-AR signaling on monocyte/macrophage activation are often anti-inflammatory and immunosuppressive [15,16,17,18,19,20,21], although under certain conditions, they can result in pro-inflammatory effects [13,22,23]. 

Β-adrenergic receptors are G-protein-coupled receptors with a role in the regulation of peripheral vascular resistance, heart function, airway tone and in the regulation of the homing, proliferation or inflammatory activity of monocytes and lymphocytes [13,22,24]. These receptors are the target of β-blockers, a class of drugs which block the action of catecholamines such as noradrenaline and adrenaline. Propranolol is the prototype of β-blockers and has been widely used in preclinical studies. It shows greater affinity for β1- and β2-ARs than β3-ARs [25] and is used alone or in combination with other drugs to treat a variety of clinical conditions, such as capillary hemangioma, supraventricular arrhythmias, migraine and arterial hypertension. In a rat model of cerebral ischemia, treatment with propranolol, a non-selective β-AR antagonist, attenuated hyperglycemia, inflammation and brain injury [26]. Propranolol also displays anti-inflammatory effects in isoproterenol-stimulated microglia and macrophage cell lines [26]. An ameliorating effect of propranolol, a non-selective β-adrenoceptor antagonist, was observed in a clinical outcome of experimental autoimmune encephalomyelitis (EAE) that was evaluated in Dark Agouti (DA) rats [26], which are the most commonly used experimental animal model of multiple sclerosis. The protective effect of propranolol administration correlated with the increased proportion of anti-inflammatory CD163- and IL-10-expressing microglia, increased phagocytic capacity of the microglia [20] and the activation of nuclear factor (erythroid-derived 2)-like 2 (Nrf2)/heme oxygenase(HO)-1 axis, which regulates multiple cytoprotective responses [27].

Even though numerous lines of research have focused on determining the effect of β-adrenergic signaling on the macrophage response, to date, there is some debate over whether the β-AR modulation of macrophage function results in a pro- or anti-inflammatory response. In particular, no information is available on whether the β-adrenergic signaling induced by molecular β-AR antagonism may influence the monocyte differentiation process in macrophages and their further polarization. 

In the present study, we used molecular biology, flow cytometry and immunoenzymatic analyses to investigate whether the β-AR-blocking drug propranolol [28] was able to modulate in vitro human monocyte-to-macrophage differentiation and to further influence the activation states of polarized macrophages.

## 2. Results

### 2.1. β-Adrenoreceptor (AR) mRNA Levels Are Similar in Monocytes and Macrophage Populations

First, we analyzed β1- and β2-AR mRNA levels in monocytes and in M0, M1-like (M IFN + LPS) and M2-like (M IL-10) macrophage populations. We found that β1-AR mRNA levels did not significantly change in macrophage populations when compared to monocytes. On the contrary, β2-AR levels decreased in M1-like (M IFN + LPS) macrophage cells when compared to monocytes (Figure 1a).

Of note, within the same population, β2-AR mRNA levels were higher than β1-AR mRNA levels (Figure 1b). When we evaluated whether the treatment with propranolol altered β-AR mRNA cellular levels, we observed that the adrenergic ligand propranolol left β-AR mRNA expression unchanged in all macrophage populations.

### 2.2. Propranolol Promotes Macrophage Differentiation of In Vitro Human Monocytes towards an Anti-Inflammatory Phenotype and Influences Macrophages towards an M2-like Phenotype in Their Polarization 

To characterize monocyte-derived macrophages, we measured the expression of markers known to be M1- or M2-related [29,30]. The considered M1 markers were HLA-DR, CD16 and CCR7, whereas the M2 markers were CD163, CD206 and CD36. Dose-response experiments established that propranolol effects were dose dependent, and 1 µM was chosen as the optimal reagent concentration for the appearance of the CD16 macrophage differentiation marker (Figure 2a,b) and for the maintenance of high cell viability (Figure 2c).

We demonstrated that the untreated control M0 macrophages expressed both the M1 and M2 markers (Figure 3). 

Propranolol treatment caused an increased percentage of cells that were positive for the M2 markers CD206 and CD163 and significantly increased the cells that were positive for the M1 marker CD16 and the expression of CD36 (as mean fluorescence intensity, MFI) (Figure 3). In our in vitro experiments, we confirmed that macrophages are very plastic, because the addition of the pro-inflammatory condition IFN-ɣ plus LPS in the culture drove M0 macrophages toward an M1-like phenotype characterized by increased HLA-DR fluorescence intensity, decreased CD36 (as MFI) and an increased percentage of CD16 and CCR7 positivity (control M IFN + LPS vs. control M0) (Figure 3). Conversely, the addition of the anti-inflammatory stimulus IL-10 drove them toward an M2-like phenotype characterized by increased percentages of CD163 and CD206 positivity and decreased CD36 (as MFI) (control M IL-10 vs. control M0). 

When we analyzed the expression of surface markers in propranolol-differentiated macrophages, we observed that the β-AR blocker significantly increased the percentage of cells that were positive for CD16, CD163 and CD206 in all macrophage populations (Figure 3) and prevented upregulation of the M1 marker HLA-DR on M (IFN + LPS) macrophages. Propranolol also induced the upregulation of CCR7 in M IL-10. Concerning the M2 phenotypic markers, we observed that the β-AR ligand only increased the expression of CD163 and of CD206 (as MFI) in M IL-10 macrophages (Figure 3). Of note, propranolol was able to induce an increased expression (MFI) of CD36 in both M0 and M IL-10 populations. 

### 2.3. Propranolol Promotes Endocytosis and Attenuates Physiological ROS Generation in Macrophage Populations

When we monitored the CD206-mediated endocytic uptake of fluorescent-labeled dextran in macrophage populations by flow cytometric analysis, we found that propranolol-differentiated M0, M (IFN + LPS) and M (IL-10) populations had endocytic abilities that were higher than those observed in the relative controls (Figure 4a).

To determine ROS cellular levels, we monitored the fluorescence intensity of H2DCF-DA-incubated macrophages by flow cytometric analysis. We found that the levels of ROS were higher in both the untreated control M0 and the M (IFN + LPS) macrophages than in the M (IL-10) macrophages (Figure 4b). Propranolol treatment was able to attenuate the physiological ROS level in all macrophage populations (Figure 4b).

To address the question of whether the effects of propranolol are due to its beta-receptor antagonist activity, next, we evaluated macrophage endocytosis and physiological ROS levels in monocytes co-treated with the β-AR agonist isoproterenol and the β-AR antagonist propranolol. By coincubating monocytes with isoproterenol and propranolol during the differentiation to macrophages, propranolol-mediated endocytosis was unaffected (Figure 4c), whereas the attenuation of physiological ROS was completely reversed (Figure 4d); specifically, propranolol was more effective at promoting ROS attenuation in M0 (Ctr vs. Prop, ROS MFI: 93.8 ± 8.6 vs. 39.7 ± 4.9; *p* < 0.001) when compared with the drug combination (Ctr vs. Prop + Isop, ROS MFI: 93.8 ± 8.6 vs. 92.7± 21.4) (Figure 4d), thus suggesting that propranolol could exert a class effect, at least in part, by triggering cellular redox status via β-ARs.

### 2.4. Propranolol Induces Secretion of the Anti-Inflammatory and Regulatory Cytokine IL-10

Subsequent activation of the macrophages with LPS resulted in a significant increase in the release of the pro-inflammatory cytokines TNF-alpha, IL-12 and IL-6 and of the regulatory cytokine IL-10 in the cell supernatants of all macrophage populations, as compared with unstimulated cells (Figure 5). 

Propranolol treatment triggered statistically significant upregulation of IL-10 secretion in culture supernatants from all populations (Figure 5). The bias toward an M2-like phenotype induced on macrophages by the β-adrenergic ligand propranolol was further supported by the lack of TNF-α and IL-12p70 pro-inflammatory cytokine secretion, with levels comparable to those of controls (Figure 5).

### 2.5. β-Adrenergic Signaling Involves PPARɣ Expression

PPARɣ is a nuclear factor required to drive M2 macrophage polarization [29]. PPARɣ basal expression levels were reduced in M (IFN + LPS) macrophages when compared to M0 macrophages (Figure 6a). 

Treatment with propranolol decreased the expression of PPARɣ in both M0 and in M (IL-10) macrophages, whereas it caused an approximately 10-fold increase in PPARɣ expression in M (IFN + LPS) macrophages in comparison to basal levels (Figure 6b). 

### 2.6. Propranolol Exerts Antioxidant Activity by Promoting the Expression of NRF2 

In consideration of the pivotal role carried out by the transcription factor NRF2 and its synergistic interaction with the PPARγ pathway to promote the expression of antioxidant genes that ultimately exert anti-inflammatory functions [27], we first investigated the activation level of NRF2 in monocytes that were exposed to propranolol. The immunofluorescence images showed that at 16 h, the propranolol triggered NRF2 activation, which was fully expressed in the nuclei, differently from the NRF2 inhibitor ML385 [31] (Figure 7a).

This upregulation of NRF2 levels was associated with an increase in the expression of the NRF2 target gene NQO1. Similar results were observed in Western blotting experiments; at 16 h of exposure, propranolol increased the protein levels of both NRF2 and its target genes, NQO1 and HO-1. Next, we measured NRF2 and NQO1 levels on propranolol-differentiated M0 macrophages and on M (IFN + LPS) and M (IL-10) polarized macrophages on day 8. In our immunofluorescence analysis, we found that the expression of NRF2 resulted upregulated in M (IL-10) macrophages, whereas it resulted significantly downregulated in M0 and M (IFN + LPS) macrophages (Figure 7b). However, the downstream antioxidant enzyme of NRF2, NQO1, was maintained effectively elevated in response to propranolol (Figure 7b). 

## 3. Discussion

Macrophages are the main inflammatory cell type involved in health and disease [32,33]. The adrenergic modulation of these cells offers a way to regulate their functions, thus providing opportunities for new therapeutic approaches in cancer and many other relevant human pathologies, such as cardiovascular diseases, aging and neurodegenerative diseases.

The present study was undertaken to evaluate the impact of the β-AR antagonism of propranolol on the human monocyte-to-macrophage differentiation process and on further macrophage polarization toward M1- and M2-like phenotypes.

The main result of this study is that the modulation of β-AR signaling in monocyte/macrophage populations by propranolol promotes an M2-like macrophage phenotype. Two independent lines of evidence support this conclusion. First, we showed that a chronic exposure of monocytes to propranolol during the monocyte-to-macrophage differentiation process caused macrophage polarization towards an M2-like phenotype by increasing the M2 scavenger receptors CD206 and CD163 in all populations. Second, propranolol induced upregulation of CD36 expression, which, like CD206 and CD163, is a member of the scavenger receptor family, and it is important for macrophage foam cell formation and M2 polarization [34]. This receptor also participates in the internalization of apoptotic cells and pathogens [34,35]. Furthermore, the upregulation of scavenger receptors in response to propranolol was associated with the upregulation of CD16, a low-affinity Fc receptor for IgG antibodies. CD16 is considered an M1 marker for its involvement in the removal of immune complexes during infection. However, the phagocytosis of immune complexes as a result of FcgRI engagement of macrophages is also known to be a trigger toward the adoption of an M2-like phenotype [36] that results in a reduction in inflammatory cytokine and chemokine levels and an induction of immunomodulatory IL-10. Collectively, these results indicate that the β-AR signaling antagonism in macrophages induced by the β-AR blocker is likely to positively influence phagocytosis, thus increasing anti-inflammatory clearance activity.

Further information on macrophage phenotypes came from our analysis of the M1 macrophage activation marker HLA-DR. Here, we showed that propranolol downregulated HLA-DR in M1 conditions, an event that supports the M2-like profile generated by the β-AR signaling antagonism. Our results on the ability of propranolol to affect the M1 pro-inflammatory phenotype are consistent with finding by Guo et al., who found that propranolol inhibits the pro-inflammatory MCP-1 and CCR2 protein expression of macrophages [37]. 

Of note, in our in vitro study, macrophages exhibited significant heterogeneity, which was determined by various factors, such as the conditions of cell culture and treatment. This heterogeneity can result in variations in the expression of cell surface markers and contradictory results. For instance, there can be an overall increase in the number of cells positive for a certain marker, but the quantity of the marker per cell is reduced, which in turn can cause a decrease in the fluorescence (see Figure 2 and Figure 3). Therefore, it is essential to use other techniques in addition to flow cytometry analysis to obtain comprehensive and accurate results at the population level as well as at the single-cell level. 

As expected, propranolol increased the production of IL-10, a key anti-inflammatory and regulatory cytokine, thus confirming its ability to skew macrophages towards an M2-like anti-inflammatory phenotype.

Additional information on macrophage features was derived from the investigation of macrophage endocytic activity, ROS production and cytokine expression, features which are directly related to macrophage functions. Our study showed that propranolol increased the endocytic ability of macrophage populations. This result is in line with the increased expression of the endocytic receptor CD206 that was observed in propranolol-differentiated macrophages, thus indicating activation toward an M2-like phenotype. Macrophages produce high levels of ROS, and this feature is regarded as a hallmark of macrophage activation [38]. ROS levels are maintained at a high level in M1 macrophages, whereas M2 activation is accompanied by increased arginase-1 activity and reduced ROS levels. Accordingly, under our experimental conditions, M0 and M (IFN + LPS) macrophages showed higher basal levels of ROS than did M (IL-10) ones. Given that propranolol treatment invariably led to a decrease in ROS levels in macrophages, it implies that propranolol may promote an M2-like macrophage phenotype, at least in part, by triggering cellular redox status via β-ARs [39], which are known to be important in the regulation of macrophage ROS production [40].

To understand the molecular signature of the transcriptional mechanisms leading to the M2-like phenotype induced by β-adrenergic signaling, we analyzed the expression of the transcription factor PPARγ. We demonstrated that the signal transduction pathway involved in propranolol’s M2-promoting effects is associated with PPARɣ upregulation [29]. PPARγ activation has been reported as a hallmark of M2 macrophages [41] and a transcription factor for the expression of CD36 [42]. Our results here suggest that PPARγ activation may be responsible for the reduced expression of M1 inflammatory markers, for the upregulation of scavenger receptors and for the increased anti-inflammatory macrophage phenotypes observed in response to β-AR ligands. Several reports have shown a crucial role of NRF2 in tuning the balance of M1/M2 macrophages, thus identifying NRF2 as a molecular target to control macrophage inflammatory response by upregulating the intracellular redox metabolism [43]. The induction of HO-1 can switch macrophages from the M1 pro-inflammatory to the M2 anti-inflammatory phenotype [44,45,46] by regulating IL-10 expression [47] and IL1β [48]. Our study showed that propranolol increased the expression of NRF2 in monocytes and that this upregulation of NRF2 levels was associated with an increase in the expression of the NRF2 target gene NQO1. 

β-ARs (β1-ARs, β2-ARs and β3-ARs) belong to the GPCR superfamily and activate adenylyl cyclase. Results from previous studies show that β-ARs can interact with different transduction proteins in the cell membrane and that β-blockers like propranolol show both inverse agonism for Gs-stimulated adenylyl cyclase and partial agonism for the mitogen-activated protein kinases and extracellular signal-regulated kinase 1/2 [49]. Therefore, there is the possibility that propranolol acting as an agonist can promote the differentiation of human monocytes towards M2-like macrophages by activating the expression of NRF2- and PPAR-γ-dependent antioxidant and anti-inflammatory genes simultaneously. Further studies will need to provide insight into the network of molecules that orchestrate the regulation of macrophage biology by β-blockers.

## 4. Materials and Methods

### 4.1. Reagents

Recombinant human (rh) macrophage colony-stimulating factor (M-CSF) and rh interleukin (IL)-10 were from R&D Systems (Minneapolis, MN, USA). Interferon (IFN)-γ, anti-CD14-coated microbeads, phycoerytrin (PE)-CD163 monoclonal antibody (mAb) and fluorescein isothiocyanate (FITC)-CD206 mAb were from Miltenyi Biotec (Gladbach, Germany). Allophycocyanin (APC)-CD16, APC-Alexa Fluor 750-human leukocyte antigen-D region-related (HLA-DR) and APC-Alexa Fluor 700-CD36, PE-Cyanine dye Cy7 (PE-Cy7)-CD197 (CCR7) mAbs were from Beckman Coulter (San Jose, CA, USA). PE-CD1a and FITC-CD14 mAbs were from PharMingen (San Diego, CA, USA). Fetal bovine serum (FBS) was from Hyclone Laboratories (Logan, UT, USA). RPMI 1640 was from GIBCO (Paisley, UK). Sytox^®^ Blue dead cell stain and 2-7-dichlorodihydrofluorescein diacetate (H2DCF-DA) were purchased from Molecular Probes (Carlsband, CA, USA). Propranolol, isoproterenol, lipopolysaccharide (LPS; from Escherichia coli strain 0111:B4), trypan blue solution and FITC-dextran were from Sigma-Aldrich (Milan, Italy). ML385, a novel and specific Nrf2 inhibitor, was purchased from Selleck Chemicals (Cologne, Germany).

### 4.2. In Vitro Cell Cultures

Peripheral blood mononuclear cells (PBMCs) were isolated from buffy coats obtained from healthy blood donors. The use of blood from this source was exempt from our institutional review board. PBMCs were incubated with anti-CD14-coated microbeads (Miltenyi Biotec), and monocytes were sorted with the MiniMacs Separation Unit (Miltenyi Biotec) magnetic device, according to the manufacturer’s instructions. Monocyte-derived macrophages (termed M0 macrophages) were obtained by culturing adherent monocytes for 6 days in complete medium [RPMI 1640 supplemented with 1% nonessential amino acids, 1% sodium pyruvate, 10,000 U/mL penicillin-streptomycin (Gibco, Karlsruhe, Germany), 5 × 10^−5^ M 2-mercaptoethanol (Merck, Darmstadt, Germany) and 10% fetal bovine serum (Hyclone Laboratories, Logan, UT, USA)] supplemented with 10 ng/mL recombinant human (rh) macrophage colony-stimulating factor (M-CSF). M0 macrophages were then polarized towards M1-like phenotype by adding 10 ng/mL rh interferon (IFN)-γ and 10 ng/mL LPS [M (IFN + LPS) macrophages] or towards M2-like phenotype by adding 10 ng/mL rh interleukin (IL)-10 [M (IL-10)] for 18 h [29,50,51]. Macrophages were washed with warm phosphate-buffered saline (PBS) and harvested using TrypLETM Express Enzyme (Gibco, Karlsruhe, Germany). Trypan blue exclusion assay (Sigma-Aldrich) and light microscope (Nikon Eclipse Ni-U, Nikon Corporation, Tokyo, Japan) were used to assess cell viability and cell morphology, respectively. Propranolol, ranging from 0.1 to 30 μM, was added to monocyte cultures at time 0 and at day 3. 

### 4.3. RNA Isolation and Quantification

Total RNA was extracted from human cells using TRIzol Plus RNA Purification Kit (AMBION by Life, Technologies, Carlsband, CA, USA). The concentration and purity of the RNA solution was analyzed using the Agilent 2100 bioanalyzer with an RNA LabChip (RNA 6000 Nano kit, Agilent, Milan, Italy). Total RNA was retrotranscribed using the High-Capacity RNA-to-cDNA Kit (Applied Biosystems Carlsbad, CA, USA). Real-time PCR reactions were carried out in 96-well plates using cDNA and TaqMan gene expression assays (Applied Biosystems, Carlsbad, CA, USA) which include specific primers and fluorescent probes for the following genes: β1-ARS (ADRB1, Hs 02330048_s1), β2-ARS (ADRB2, Hs00240532_s1), NOS2 (Hs01075529_m1), PPARɣ (Hs01115513_m1) and GAPDH (Hs 99999905_m1). Real-time PCR was performed on ABI Prism 7500 Fast Sequence Detector (Applied Biosystems). 

### 4.4. Flow Cytometric Analysis of Macrophage Phenotypes and Endocytosis

Phenotypic surface markers were determined by staining macrophages for 30 min at 4 °C with the indicated monoclonal antibodies. To assess mannose receptor-mediated endocytosis, macrophages (1 × 10^6^ cells/mL) were incubated with FITC-dextran (1 mg/mL; Sigma-Aldrich) for 45 min at 37 °C. The percentage and the mean fluorescence intensity (MFI) of FITC-positive cells were determined to assess cell internalization ability. To exclude dead cells from analysis, 1 μM Sytox Blue nucleic acid staining (Molecular Probes, Carlsband, CA, USA) was used. A total of 10,000 cells were acquired by Gallios Flow cytometer and equipped with 3 lasers (405 nm, 488 nm, 633 nm, Beckman Coulter), and data were analyzed with Kaluza Analysis Software v. 2.1 (Beckman Coulter). 

### 4.5. Flow Cytometric Analysis of Reactive Oxygen Species Production

Monocyte-derived macrophages were incubated (1 × 10^6^ cells/mL) with H2DCF-DA at a final concentration of 2.5 μM to monitor the production of reactive oxygen species (ROS). After 45 min of incubation in the dark at 37 °C, cells were washed twice with ice-cold PBS/FCS and 2′,7′-dichlorofluorescein (DCF), and mean fluorescence intensity was measured in FL-1 by flow cytometry (Gallios Flow Cytometer; Beckman Coulter) with an excitation wavelength of 488 nm and an emission wavelength of 530 nm.

### 4.6. Cytokine Secretion Analysis in Macrophage Culture Supernatants

After 20 h, supernatants were collected from cultures of human macrophages (7 × 10^5^ cells per mL) treated or left untreated on 24-well plates, centrifuged and stored at −80 °C. The concentrations of IL-12 p70, TNF-alpha, IL-6 and IL-10 were determined by enzyme-linked immunosorbent assay (ELISA; BD OptEIA™ Kits; BD Biosciences, San Di-ego, CA, USA) following the manufacturer’s instructions. The limits of detection were as follows: IL-12p70, 7.8 pg/mL; TNF-alpha and IL-10, 16 pg/mL; IL-6, 2.2 pg/mL.

### 4.7. Indirect Immunofluorescence Labeling

For immunofluorescence microscopy, cells were plated and permeabilized as previously described [50]. Macrophages were then washed with PBST, blocked with 0.1 M glycine in PBS for 20 min and incubated with the primary antibodies, mouse anti-NAD(P)H:quinone oxidoreductase 1 (NQO1) (1:500; Abcam, Cambridge, UK) and rabbit Anti-NRF2 polyclonal antibody (1:500; PA5-27882, Thermo Fisher Scientific, Waltham, MA, USA), in PBS at 37 °C for 90 min. After washing with PBST, cells were incubated at 37 °C for 30 min with the following fluorescence-labeled secondary antibodies: Alexa Fluor 488 goat anti-rabbit (Thermo Fisher Scientific) or Alexa Fluor 594 goat anti-mouse at a 1:100 dilution (Invitrogen, Carlsbad, CA, USA). Cells were then incubated with DAPI (Thermo Fisher Scientific) to stain the nucleus, and images were obtained using an upright microscope (Nikon Eclipse Ni-U, 60× magnification, Nikon Corporation, Tokyo, Japan). Nikon NIS-Elements 5.0 imaging software (Nikon Corporation) was used for collecting images and for post-acquisition processing. The analysis was performed in triplicate from three different fields.

### 4.8. Statistical Analysis

Mean values and standard deviations (SDs) were calculated for each variable under study. Statistical analysis was performed by GraphPad Prism 5 software (San Diego, CA, USA). Normally distributed data were analyzed using one-way ANOVA with a Bonferroni post hoc test to evaluate the statistical significance of intergroup differences. Significant changes in gene expression were determined using unpaired two-tailed Student’s *t*-tests on delta Ct individual values according to Schmittgen and Livak [52]. *p* values < 0.05 were considered statistically significant.

## 5. Conclusions

Our in vitro findings also call for in vivo studies to verify the potential significance of our findings for humans. Along with the current knowledge, our results show that propranolol, by modulating β-AR signaling, causes macrophages to switch towards an anti-inflammatory and reparative M2-like phenotype. The ability of propranolol to switch macrophages into an M2-like phenotype and to increase the anti-inflammatory clearance activity of macrophages might be a key mechanism to explain, in part, the increased survival associated with β-AR blocker-based therapies for patients with cancer [53,54], previous myocardial infarction and atherosclerosis [55,56]. In ischemic hearts, β-blockers are thought to be beneficial due to their influence in reducing cardiac work and limiting cardiomyocyte death; however, previous research also has demonstrated that they can regulate the magnitude of innate immune responses following cardiac injury [48].

## Figures and Tables

**Figure 1 ijms-25-03683-f001:**
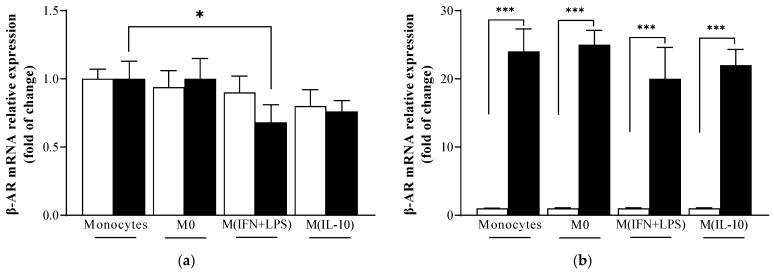
Expression levels of β1- and β2- adrenergic receptors (ARs) in human monocyte, M0, M (IFN + LPS) and M (IL-10) macrophage populations evaluated by quantitative real-time PCR. (**a**) (≤☐) β1-AR mRNA levels did not significantly differ in macrophage populations when compared to monocytes that were chosen as controls, with mean of 2^−ΔΔCt^ = 1. On the contrary, (■) β2-AR mRNA levels decreased in M1 (IFN + LPS) macrophages when compared to monocytes; (**b**) shows the expression differences between (≤) β1- (chosen as control with mean of 2^−ΔΔCt^ = 1) and (■) β2-ARs in monocytes, M0, M (INF + LPS) and M (IL-10) macrophages. (■) β2-AR mRNA levels were higher than (≤) β1-AR mRNA levels; data shown are mean (±SE). Significant differences are indicated by *p*-values: * *p* < 0.05; *** *p* < 0.001.

**Figure 2 ijms-25-03683-f002:**
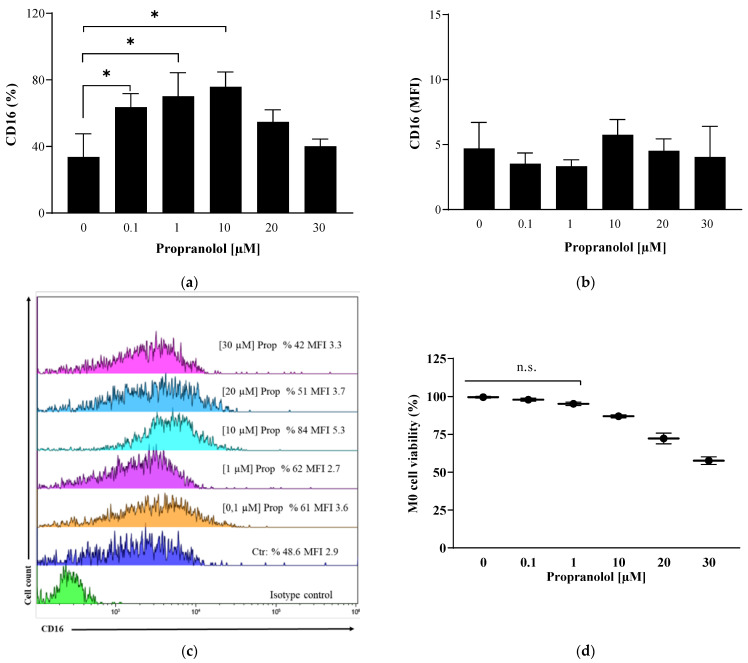
Dose-dependent effects of propranolol during monocyte-to-macrophage differentiation. (**a**) Flow cytometric analysis of surface CD16 marker expression on M0 macrophages. Histograms show the percentages of positive cells (%) and (**b**) the mean fluorescence intensity (MFI); (**c**) a 2-D plot image representative of the fluorescence intensity of CD16 in M0 macrophages; (**d**) dose–response curve of cell viability: M0 macrophages were exposed to 0.2% trypan blue and then counted in a hemocytometer to calculate the percentage of dead cells. Results are expressed as mean value ± SD of 3 independent experiments. Significance was determined by one-way ANOVA followed by Tukey’s post hoc analysis; not significance (n.s.); * *p* < 0.033.

**Figure 3 ijms-25-03683-f003:**
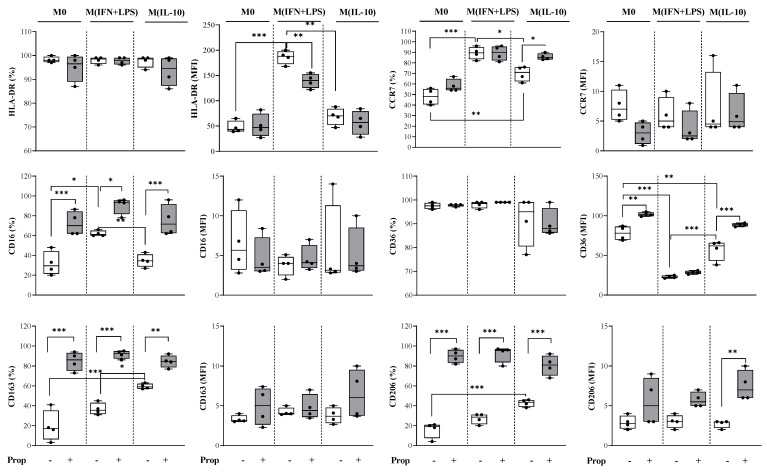
Analysis of surface marker expression on macrophage populations differentiated with propranolol and further activated toward an M1-like phenotype or an M2-like phenotype. Flow cytometric analysis of M1 and M2 surface marker expression in M-CSF-differentiated M0 macrophages and in activated M (IFN + LPS) and M (IL-10) macrophages. Monocytes were induced to differentiate into M0 macrophages in medium supplemented with 10 ng/mL (rh)M-CSF. On day 7, cells were further stimulated for 24 h by the addition of 10 ng/mL LPS plus 10 ng/mL IFN-ɣ for M (IFN + LPS) macrophages, or 10 ng/mL IL-10 for M (IL-10) macrophages. The non-selective β-AR antagonist, propranolol (Prop; 1 µM), was added to monocyte cultures at time 0 and at day 3. Cells were harvested and analyzed by Gallios Flow Cytometer, and the results were analyzed by using FACS Kaluza analysis 2.1 software (Beckman Coulter). M0 macrophages expressed both M1 and M2 markers. Propranolol modulated in vitro human monocyte-to-macrophage differentiation towards macrophages with an anti-inflammatory phenotype. Results are expressed as percentages of positive cells and mean fluorescence intensity (MFI) (mean ± SD; n = 4). Significant differences are indicated by *p*-values: * *p* < 0.033; ** *p* < 0.002; *** *p* < 0.001.

**Figure 4 ijms-25-03683-f004:**
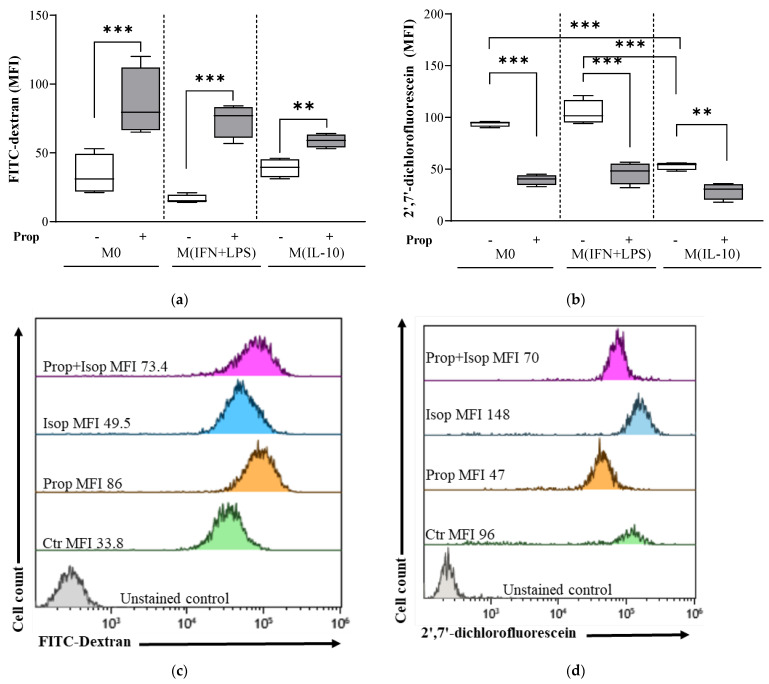
Analysis of macrophage endocytosis and physiological reactive oxygen species (ROS) levels of the distinct macrophage populations. Flow cytometric analysis of cellular uptake of FITC-dextran or H2DCF-DA by M0, M (IFN + LPS) and M (IL-10) macrophage populations. Propranolol (Prop; 1 µM) was added at time 0 and day 3. In 8 days, cells were harvested, supplemented with FITC-dextran and analyzed by flow cytometry. (**a**) Propranolol-differentiated M0, M (IFN + LPS) and M (IL-10) populations had endocytic abilities higher than those observed in the relative controls. (**b**) Propranolol treatment was able to attenuate the physiological ROS levels in all macrophage populations. Similar experiments were repeated in M0 macrophages differentiated in the presence or absence of β-AR agonist isoproterenol (Isop; 1 µM) and β-AR antagonist Prop (1 µM) and added to monocytes at time 0 and at day 3. Two-dimensional plot images representative of cellular uptake of (**c**) FITC-dextran and (**d**) H2DCF. Results are expressed as a mean of the median fluorescence intensity (MFI) (mean ± SD; n = 4). Significant differences are indicated by *p*-values: ** *p* < 0.002; *** *p* < 0.001.

**Figure 5 ijms-25-03683-f005:**
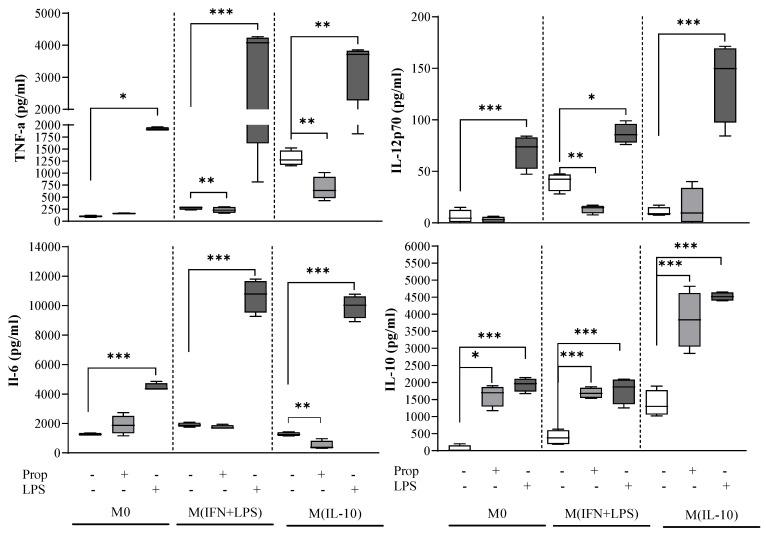
Analysis of cytokine production in macrophage populations. Macrophage culture supernatants were collected at the end of day 8. Levels of IL-12p70, IL-6, TNF-alpha and IL-10 were determined by ELISA. LPS (100 ng/mL) was used as positive control stimulus for pro- and anti-inflammatory cytokine production. Propranolol (Prop; 1 µM) triggered statistically significant upregulation of production of the regulatory cytokine IL-10. Results are expressed as mean value ± SD of 4 independent experiments. Significance was determined by one-way ANOVA followed by Tukey’s post hoc analysis; * *p* < 0.033; ** *p* < 0.002; *** *p* < 0.001.

**Figure 6 ijms-25-03683-f006:**
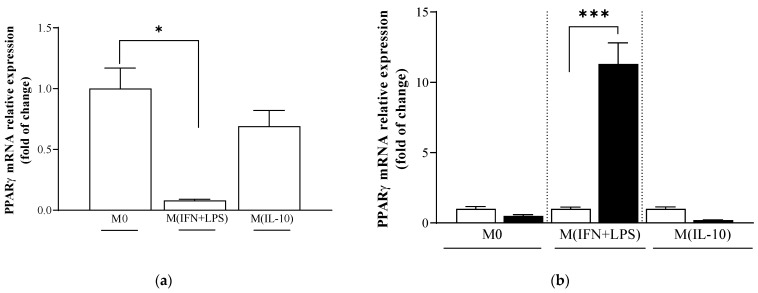
Analysis of PPARɣ gene expression in macrophage populations. mRNA PPARɣ transcript levels were assayed by quantitative real-time PCR. (**a**) (☐) PPARɣ basal expression levels were reduced in M (IFN + LPS) macrophages when compared to M0 macrophages. (**b**) (■) Propranolol increased PPARɣ mRNA levels in M (IFN + LPS) macrophages but reduced its levels in M0 and M (IL-10) macrophages. Results are expressed as mean value ± SD of 3 independent experiments. Significance was determined by one-way ANOVA followed by Tukey’s post hoc analysis; * *p* < 0.033; *** *p* < 0.001.

**Figure 7 ijms-25-03683-f007:**
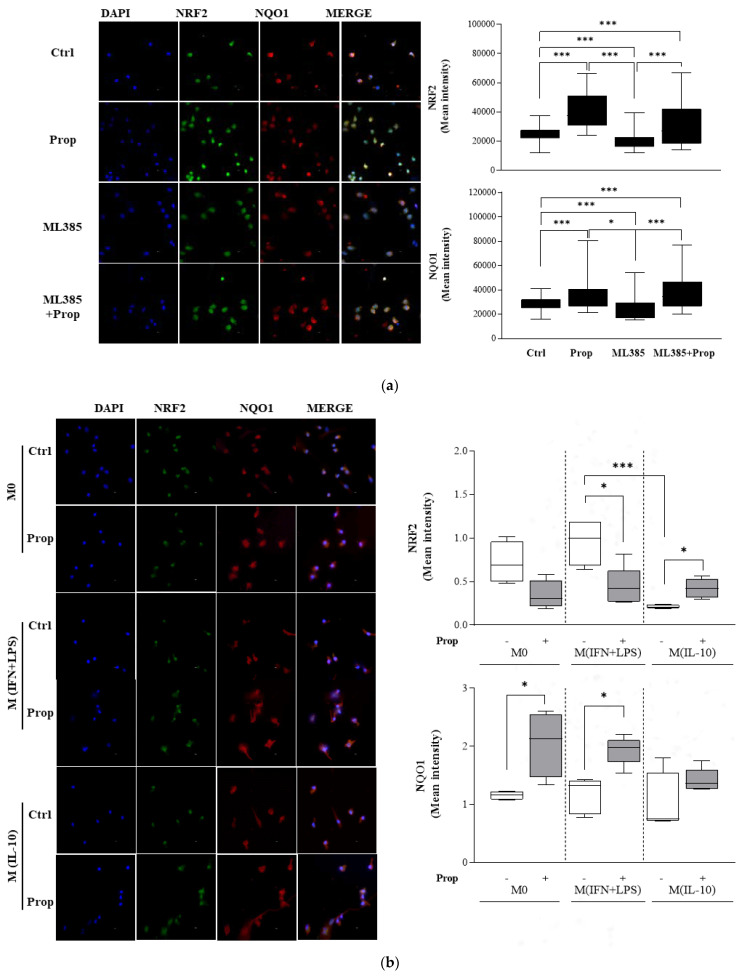
Immunofluorescence analysis of NRF-2 and NQO1 expression in monocytes and macrophage populations pretreated for 30 min with 5 µM ML385 and further treated with propranolol (Prop; 1 µM) for 16 h. Representative immunofluorescent staining of NRF2 and NQO1 (**a**) in monocytes and (**b**) in macrophages, with relative quantification of the intensity of the fluorescence signal for positive cells. Immunofluorescence analysis was performed using the ImageJ 1.52a software. DAPI was used to counterstain the nuclei. Data are presented as the mean ± SD for each group (N = 3). Significance was determined by one-way ANOVA followed by Tukey’s post hoc analysis; * *p* < 0.033; *** *p* < 0.001.

## Data Availability

The data presented in this study are contained within the article. Individual values are available from the corresponding authors upon reasonable request.

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
