# Peer review of "Propranolol Promotes Monocyte-to-Macrophage Differentiation and Enhances Macrophage Anti-Inflammatory and Antioxidant Activities by NRF2 Activation"

_ijms, 2024, doi:10.3390/ijms25073683_

Round 1

Reviewer 1 Report

Comments and Suggestions for Authors

The manuscript presents the effects of propranolol, an unspecific beta-receptor blocker on human monocytes differentiating into M1 or M2 macrophages. In the title of the manuscript, the authors suggest that this compound promotes the differentiation of monocytes into M2 macrophages. As currently presented, the data do not sustain this statement. The classical bipolar view of monocyte differentiation into either M1 or M2 macrophages is derived from the (pre genomic) T helper 1 resp. T helper 2 concept and needs certainly a revision in the post genomic era (see eg. for review Martinez & Gordon 2014). Actually, the authors show in Figs 2 and 3 that both M1 markers such as CD16 and M2 markers such as CD 206 are enhanced in the cell population treated with propranolol. Does this compound simply promote the monocyte-macrophage differentiation in general?

There is another problem concerning the flow cytometry data presented in figs 2 and 3. The authors give percentage values of positive cells as well as mean fluorescence identities. Both datasets give conflicting or even contradictory results with all markers analyzed. How is this possible? How many cells were gated? Was there a compensation problem? Here, 2-D-plots representing the macrophage subpopulations as defined by the specific markers would be necessary.

The data concerning the physiological responses of treated vs. untreated macrophages are more consistent and clearly merit publication. There are only two points to consider. First, the authors show that treated macrophages secrete IL-10 in the same order of magnitude as in the IL-10 treatment group (Fig. 5). How relevant are the data of the corresponding group given the fact that the cells were treated with 10 ng/ml (see legend of fig. 3) and that the amounts measured by ELISA in the supernatants of the same cells are below 10 ng?

The second point is that the propranolol effects are considered as due to its beta-receptor antagonist activity. To enhance belief in this hypothesis, experiments with the most important results (see e.g. fig. 4) should be repeated in the presence of a beta-receptor agonist. In the case of the suggested beta-2-receptors (see fig. 1; could be presented as a table), clenbuterol would be an option.

Taken together, although the present manuscript clearly has some merits, it needs a thorough reworking before being acceptable for publication.

Comments on the Quality of English Language

English is not a major issue.

Author Response

2. Point-by-point response to Comments and Suggestions for Authors Reviewer 1

Comments 2.1: The manuscript presents the effects of propranolol, an unspecific beta-receptor blocker on human monocytes differentiating into M1 or M2 macrophages. In the title of the manuscript, the authors suggest that this compound promotes the differentiation of monocytes into M2 macrophages. As currently presented, the data do not sustain this statement. The classical bipolar view of monocyte differentiation into either M1 or M2 macrophages is derived from the (pre genomic) T helper 1 resp. T helper 2 concept and needs certainly a revision in the post genomic era (see eg. for review Martinez & Gordon 2014). Actually, the authors show in Figs 2 and 3 that both M1 markers such as CD16 and M2 markers such as CD 206 are enhanced in the cell population treated with propranolol. Does this compound simply promote the monocyte-macrophage differentiation in general?

Response 2.1: We understand the reviewer's criticism about the title, and we agree that it needed some tweaking. We have taken the reviewer’s comments on board and rephrased it to better reflect the content. Thanks for your input.

Comments 2.2: There is another problem concerning the flow cytometry data presented in figs 2 and 3. The authors give percentage values of positive cells as well as mean fluorescence identities. Both datasets give conflicting or even contradictory results with all markers analyzed. How is this possible? How many cells were gated? Was there a compensation problem? Here, 2-D-plots representing the macrophage subpopulations as defined by the specific markers would be necessary.

Response 2.2:

As the reviewer correctly pointed out, the datasets give contradictory results. However, we have addressed this issue by analyzing the FACS data using a 2D-plot. In this plot, control M0 macrophages are CD16-negative or weakly positive, with minimal fluorescence, and CD16-positive, with high fluorescence. When treated with 1 µM propranolol, some CD16-negative/weakly positive cells become CD16-positive, but with a lower mean fluorescence intensity (MFI) than the starting CD16-positive cells. This results in a reduction of the average amount of fluorescence in the total CD16-positive cells. This new figure is now presented as Figure 2c.

The marked heterogeneity of macrophages resulting from their specific cell culture and treatment conditions may cause an increase in cell percentage that does not correlate with an increase in the MFI, despite we regularly gated cells based on physical parameters, and excluded dead cells, debris, or cells with higher size/granularity from the analysis. Additionally, we have applied the same compensation to the acquired file data. We have included the number of cells acquired by FACS in the Methods section of the revised manuscript (Page 15). The number of cells gated for FACS analysis varied depending on the experiment and the specific population being analyzed. We followed standard practice for gating and excluding dead cells, debris, or cells with higher size/granularity from the analysis. Although improving the accuracy of FACS data is important, we believe that the variability observed in the MFI data does not significantly affect the outcomes of this study but it needs to be taken into account. Therefore, we have provided additional commentary on these results in the Discussion section of the revised manuscript (Page 13).

Comments 2.3: The data concerning the physiological responses of treated vs. untreated macrophages are more consistent and clearly merit publication. There are only two points to consider. First, the authors show that treated macrophages secrete IL-10 in the same order of magnitude as in the IL-10 treatment group (Fig. 5). How relevant are the data of the corresponding group given the fact that the cells were treated with 10 ng/ml (see legend of fig. 3) and that the amounts measured by ELISA in the supernatants of the same cells are below 10 ng?

Response 2.3: We understand the reviewer's concern about the low IL-10 levels measured by ELISA in the supernatants of M (IL-10) macrophages, which were below the concentration (10 ng/ml) used for macrophage stimulation. However, it is important to consider that this could be attributed to the short lifespan of IL-10 (the mean terminal phase half-life of recombinant human IL-10 is 2.7 to 4.5 hours; PMID: 9284853), as well as the molecular mechanism that regulates the availability and action of endogenous IL-10.

Comments 2.4: The second point is that the propranolol effects are considered as due to its beta-receptor antagonist activity. To enhance belief in this hypothesis, experiments with the most important results (see e.g. fig. 4) should be repeated in the presence of a beta-receptor agonist. In the case of the suggested beta-2-receptors (see fig. 1; could be presented as a table), clenbuterol would be an option.

Response 2.4: Thank you for your important comment. Based on the suggestion of the Reviewer, we have included a representative 2D plot in the revised manuscript for the FACS analysis of endocytosis and physiological ROS conducted in the presence of 1 µM of the β-AR agonist isoproterenol (Isop). This new figure is now presented as Figure 4c-d. We have also provided additional commentary on these results in the Results section of the revised manuscript (Page 8).

Comments 2.5: Taken together, although the present manuscript clearly has some merits, it needs a thorough reworking before being acceptable for publication.

Response 2.5: We thank the Reviewer for their time and effort in evaluating our manuscript          

Reviewer 2 Report

Comments and Suggestions for Authors

Macrophages play a vital role in the human immune system and are of key importance in the pathogenesis of various diseases. One of the distinguishing features of macrophages is their plasticity. Macrophages are able to switch to both a pro-inflammatory and anti-inflammatory phenotype. A very important task for the treatment of many diseases of an immune nature is to find the ability to control this function of macrophages. In their paper, “Propranolol regulates macrophage differentiation and polarization into M2-like macrophages and enhances their anti-inflammatory and antioxidant activities by NRF2 activation,” the authors demonstrated that propranolol can regulate macrophage differentiation toward an anti-inflammatory-like phenotype. I found this work very informative and interesting, however it has a few shortcomings, which I have written about below.

Plagiarism

The Materials and Methods section is heavily overloaded with borrowed text. Please review the file attached below and correct the text. 20% of your text overlaps with this publication https://www.mdpi.com/1422-0067/23/21/13009.

Since most of the borrowed text is concentrated in the Materials and Methods section, I do not find this to be a big problem. However, I strongly recommend changing it.

Introduction

Introduction section too short.

From this introduction, I did not understand why propranolol was the drug of choice for the trial, since in addition to it there are other non-selective beta-blockers. Especially considering the fact that this medicine has many contraindications and side effects.

Please describe in more detail M2 macrophages and their influence in various diseases

General recommendation: Please expand your Introduction section by at least two paragraphs. Everything I wrote above interests me directly. It is not necessary to write this exact information. You can expand the Introduction section with the information you find necessary.

Results

I don't quite understand the difference between Figures 1a and 1b. These figures have the same captions on both axes. In the text, you write the following:

1) We found that β1- and β2-AR mRNA levels did not significantly differ among monocytes and macrophage populations (Figure 1a).  Lines 83-85

2) Of note, β2-AR mRNA levels were higher than β1-AR mRNA levels in all the cell populations (Fig. 1b). Lines 93-94

Perhaps the text of paragraph 2.1 is missing some important information? Because I don't understand why in one graph the expression of β1- and β2-AR mRNA is not different, but in the other the expression of β2-AR mRNA is significantly higher for all cells, although no additional information is provided.

In addition, can you please write the units of measurement that you used for the y-axis in Figure 1a and 1b.

Despite the comments above, I find this article extremely fascinating and useful. I wish the authors of this manuscript success with their publication and future work.

Author Response

3. Point-by-point response to Comments and Suggestions for Authors Reviewer 2

Comments 3.1: Macrophages play a vital role in the human immune system and are of key importance in the pathogenesis of various diseases. One of the distinguishing features of macrophages is their plasticity. Macrophages are able to switch to both a pro-inflammatory and anti-inflammatory phenotype. A very important task for the treatment of many diseases of an immune nature is to find the ability to control this function of macrophages. In their paper, “Propranolol regulates macrophage differentiation and polarization into M2-like macrophages and enhances their anti-inflammatory and antioxidant activities by NRF2 activation,” the authors demonstrated that propranolol can regulate macrophage differentiation toward an anti-inflammatory-like phenotype. I found this work very informative and interesting, however it has a few shortcomings, which I have written about below.

Response 3.1: We thank the Reviewer for their time and effort in evaluating our manuscript.

Comments 3.2: Plagiarism

The Materials and Methods section is heavily overloaded with borrowed text. Please review the file attached below and correct the text. 20% of your text overlaps with this publication https://www.mdpi.com/1422-0067/23/21/13009.

Since most of the borrowed text is concentrated in the Materials and Methods section, I do not find this to be a big problem. However, I strongly recommend changing it.

Response 3.2: Agree. We have revised the Materials and Methods section to modify the borrowed text. Particularly, we have updated the following paragraphs accordingly: 4.2, 4.4, 4.6, 4.7, and 4.8 (Pages 14-16).

Comments 3.3: Introduction

Introduction section too short.

Please describe in more detail M2 macrophages and their influence in various diseases.

General recommendation: Please expand your Introduction section by at least two paragraphs. Everything I wrote above interests me directly. It is not necessary to write this exact information. You can expand the Introduction section with the information you find necessary.

Response 3.3: Upon the given suggestion, we have expanded the Introduction section and added additional information on propranolol and M2 macrophages to the revised manuscript (Pages 2-3). (Page 3 lines 87-99 and page 2 lines 44-72, respectively).  

Comments 3.4: From this introduction, I did not understand why propranolol was the drug of choice for the trial, since in addition to it there are other non-selective beta-blockers. Especially considering the fact that this medicine has many contraindications and side effects.

Response 3.4: Propranolol is a well-known beta-blocker that has been used for decades in the treatment of hypertension, cardiac arrhythmias, and other cardiovascular conditions. However, recent studies have shown that propranolol has potential therapeutic effects beyond its traditional use. For example, it has been shown to have anti-tumor and anti-inflammatory effects, making it a promising candidate for repurposing in the treatment of cancer and other diseases (PMID: 20615772; 22006582; 24389287). In addition, propranolol has been shown to have immunomodulatory effects, which could make it useful in the treatment of autoimmune diseases and other conditions that involve dysregulation of the immune system (PMID: 31689515). At the doses used clinically, the most common adverse effects of propranolol as well as other non-selective β-blockers arise as pharmacological consequences of β-adrenergic receptor blockade. Some of this information has been included in the revised manuscript (Pages 3-4).

Comments 3.5: Results

I don't quite understand the difference between Figures 1a and 1b. These figures have the same captions on both axes. In the text, you write the following:

1) We found that β1- and β2-AR mRNA levels did not significantly differ among monocytes and macrophage populations (Figure 1a).  Lines 83-85

2) Of note, β2-AR mRNA levels were higher than β1-AR mRNA levels in all the cell populations (Fig. 1b). Lines 93-94

Perhaps the text of paragraph 2.1 is missing some important information? Because I don't understand why in one graph the expression of β1- and β2-AR mRNA is not different, but in the other the expression of β2-AR mRNA is significantly higher for all cells, although no additional information is provided.

In addition, can you please write the units of measurement that you used for the y-axis in Figure 1a and 1b.

Response 3.5: Thank you for your important comment. We apologize for unclear text provided. Figure 1a shows the changes in expression of β1 and β2-adrenergic receptors in M0, M(INF+LPS) and M(IL-10) macrophages compared to monocytes that were chosen as control with mean 2^-ΔΔCt=1. Instead, Figure 1b shows the expression differences between β1 (chosen as control with mean 2^-ΔΔCt=1) and β2-adrenergic receptors in monocytes, M0, M(INF+LPS) and M(IL-10) macrophages. The updated text, highlighted in red, can be found on pages 3 and 4 (lines 122-136).

Round 2

Reviewer 1 Report

Comments and Suggestions for Authors

The authors did a good job in reviewing the present manuscript. Although there are still some open questions regarding the macrophage subpopulations, additional experiments have been performed, and the MS has been restructured in a way that facilitates reproduction of the results by other groups. It clearly merits publication.

Reviewer 2 Report

Comments and Suggestions for Authors

Thanks for your reply. I have no further comments on this manuscript.